# Physical and environmental drivers of Paleozoic tetrapod dispersal across Pangaea

Neil Brocklehurst[1,2], Emma M. Dunne[3], Daniel D. Cashmore[3] & Jörg Fröbisch[2,4]

The Carboniferous and Permian were crucial intervals in the establishment of terrestrial ecosystems, which occurred alongside substantial environmental and climate changes throughout the globe, as well as the final assembly of the supercontinent of Pangaea. The influence of these changes on tetrapod biogeography is highly contentious, with some authors suggesting a cosmopolitan fauna resulting from a lack of barriers, and some identifying provincialism. Here we carry out a detailed historical biogeographic analysis of late Paleozoic tetrapods to study the patterns of dispersal and vicariance. A likelihood-based approach to infer ancestral areas is combined with stochastic mapping to assess rates of vicariance and dispersal. Both the late Carboniferous and the end-Guadalupian are characterised by a decrease in dispersal and a vicariance peak in amniotes and amphibians. The first of these shifts is attributed to orogenic activity, the second to increasing climate heterogeneity.

[1] Department of Earth Sciences, University of Oxford, South Parks Road, Oxford OX1 3AN, UK. [2] Museum für Naturkunde, Leibniz-Institut für Evolutions- und Biodiversitätsforschung, Invalidenstraße 43, 10115 Berlin, Germany. [3] School of Geography, Earth and Environmental Sciences, University of Birmingham, Birmingham B15 2TT, UK. [4] Institut für Biologie, Humboldt-Universität zu Berlin, Invalidenstraße 42, Berlin 10115, Germany. Correspondence and requests for materials should be addressed to N.B. (email: neil.brocklehurst@earth.ox.ac.uk)

A major transition in the history of terrestrial faunas occurred during the late Paleozoic. During the Carboniferous, the tetrapod lineage diversified and evolved a fully terrestrial lifestyle with the appearance of the amniotic egg[1]. During the Permian, terrestrial ecosystems developed a more modern trophic structure, with large numbers of terrestrial herbivorous vertebrates supporting a small number of macro-carnivores[2]. These changes occurred alongside substantial environmental and climate changes throughout the globe.

During the Carboniferous, the assembly of the supercontinent Pangaea was completed. With all continents connected into a single landmass, it has often been suggested that there were few barriers to dispersal, producing a strongly cosmopolitan fauna, at least at the family-level[2–11]. Even though in more recent years provincialism and faunal variation according to palaeolatitude has been identified[12–21] there has been very little study into the historical biogeography of terrestrial vertebrates during the Paleozoic.

Many discussions of patterns of dispersal and vicariance in Paleozoic tetrapods are short notes embedded in studies whose primary focus is anatomical descriptions of species[22–25]. Such discussions lack the application of quantitative methods and instead are limited to visual examinations of a phylogenetic tree, often just presenting the biogeographic area of the basalmost member of a particular clade as the place of origin of that clade. These hypotheses are problematic, because (1) they only take into account one lineage, (2) they assume a narrow ancestral area, (3) they assume dispersal as the driving force behind the distribution patterns in clades, making no allowance for vicariance and or the possibility of climatic barriers separating populations within a species and (4) they are at risk of changes in interpretation every time a new, more basal taxon is discovered.

A global study of cosmopolitanism, vicariance and dispersal within the united supercontinent of Pangaea is integral to the understanding of early amniote evolution and diversification, providing vital information on the impact of physical and climatic barriers on different clades evolving within a single landmass. Quantitative methods, particularly event-based methods incorporating phylogenetic hypotheses, will provide a more rigorous analysis of these issues than has previously been applied to Paleozoic tetrapods.

Here, we present an examination of the patterns of dispersal and vicariance of tetrapods across Pangaea during the Carboniferous and Permian. A supertree of Carboniferous-Permian tetrapods was generated and subjected to likelihood-based biogeographic modelling analysis in order to infer the ancestral ranges of internal nodes. A stochastic mapping approach, incorporating null model generation, is used to infer rates of dispersal and vicariance through time. Increases in vicariance and decreases in dispersal rates are found in the late Carboniferous and the end-Guadalupian. The first of these shifts is attributed to the orogenic activity, the second to increasing climate heterogeneity.

## Results and Discussion

### Dispersal and vicariance rates.
Throughout most of the Carboniferous, vicariance rates remain low and relatively constant, aside from a slight peak in the Serpukovian (Fig. 1). The largest peak in vicariance through the Carboniferous is towards the end, during the Kasimovian. While vicariance rates do fall after this, they rise again during the middle Permian, reaching a peak in the Capitanian. Both of these two peaks are followed by declines in dispersal rate (Fig. 1).

The Carboniferous peak in vicariance rate and decline in dispersal is most pronounced in amphibians (Fig. 2) and in particular lepospondyls. The vicariance peak in amniotes is relatively small (Fig. 3). Amniote dispersal rates fall at this time, but they soon recover, unlike those of amphibians which remain below 0 for the rest of the Permian (Fig. 2)

Tetrapod dispersal rates recover throughout the early Permian, reaching a peak during the early Capitanian (Fig. 1). The late Capitanian vicariance peak, and the trough in dispersal that follows, is visible in all amniote clades and also the temnospondyls (Figs. 6–7). Only one lepospondyl is present by this time (*Diplocaulus minimus* from Morocco), so vicariance is obviously not possible in this clade.

### Late Carboniferous dispersal patterns.
The dispersal and vicariance patterns of amphibians and amniotes indicate differing responses to the geological and climate changes occurring during the latter half of the Paleozoic. For much of the Carboniferous, the tropical regions were covered by the coal forests, a dense belt of tropical rainforests[26]. The general climate trend throughout the Permian was towards warming and drying[27–30], albeit punctuated by wet phases relating to the waxing and waning of the Gondwana ice sheet[31]. Gradually, the substantial polar icecap present since the Devonian disappeared, the tropical belt across the equator narrowed, and the arid zones surrounding it expanded[28,29].

At the end of the Carboniferous, during the Moscovian stage, the rainforest collapse is thought to have occurred: the coal forests covering the equatorial regions were reduced to islands of rainforest in intramontane basins of the Variscan mountains[26]

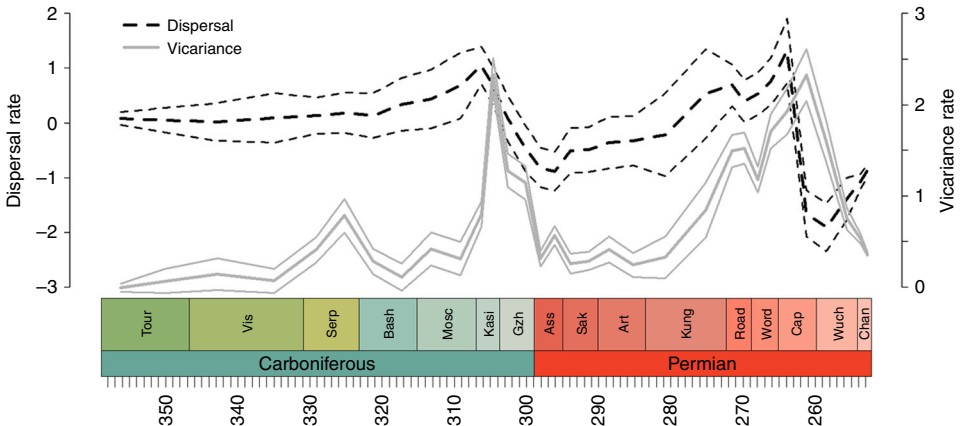

**Fig. 1** Dispersal and vicariance rates of all Tetrapoda through time. Solid grey lines represent vicariance. Dashed black lines represent dispersal. Thick lines represent the mean of the rates obtained from each stochastic mapping iteration. The thinner lines represent a standard error above and below

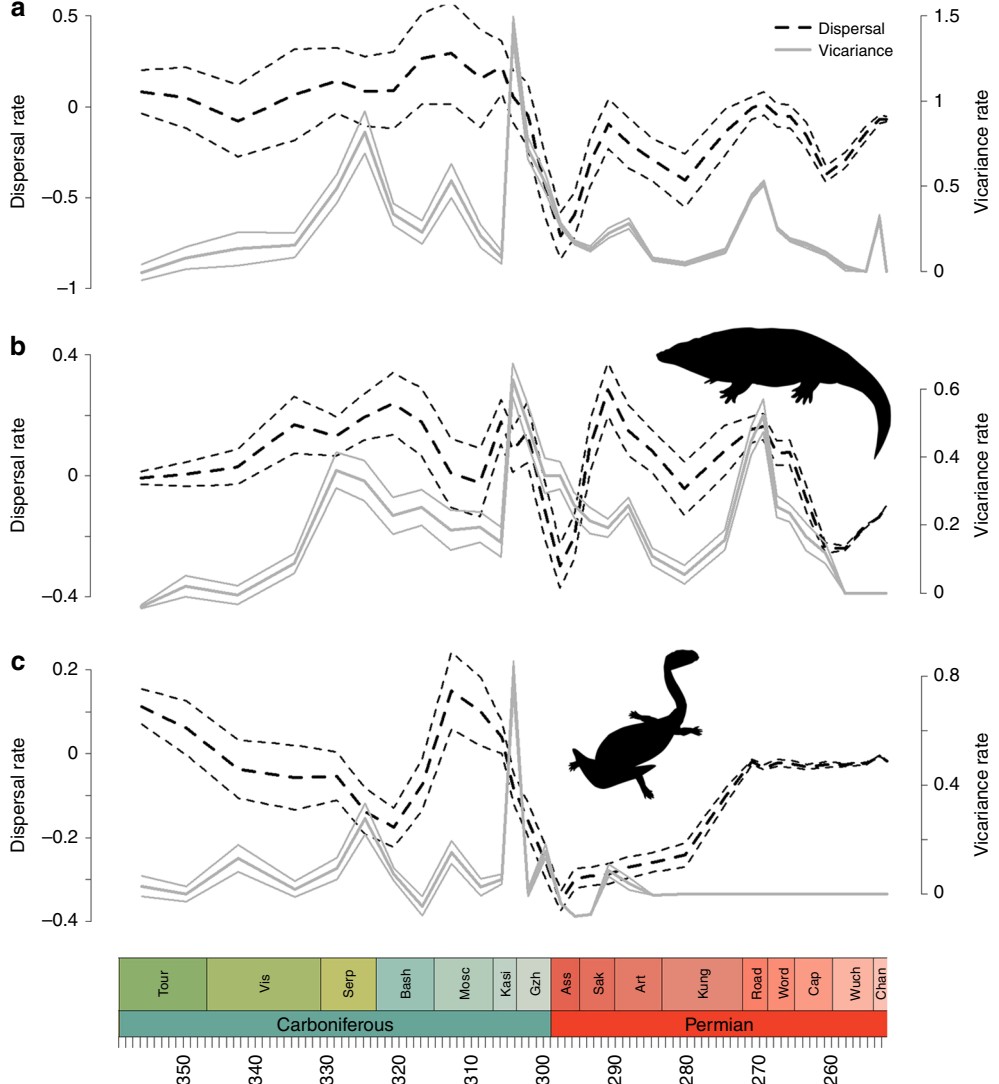

**Fig. 2** Dispersal and vicariance rates of amphibian clades through time. **a** All amphibians; **b** Temnospondyli; **c** Lepospondyli. Solid grey lines represent vicariance. Dashed black lines represent dispersal. Thick lines represent the mean of the rates obtained from each stochastic mapping iteration. The thinner lines represent a standard error above and below. Silhouettes from phylopic.org. **b** *Eryops*, by Dmitry Bogdanov (vectorized by T. Michael Keesey), available under Creative Commons Attribution-ShareAlike 3.0 Unported license. **c** *Diplocaulus*, by Gareth Monger, available under Creative Commons Attribution 3.0 Unported license

surrounded by more arid habitats[32,33]. This was accompanied by a period of mass extinction among plants[34,35] and a shift from a lycopsid-dominated flora to a fern-dominated flora[26]. The transition does not appear to have occurred at a consistent rate across the equatorial latitudes, with macrofloral and palynological evidence suggesting coal development continued into the earliest Kasimovian in American localities[26], but after this the last remnants of the lycopsid-dominated flora disappeared. Cleal et al.[26], after thoroughly reviewing the sedimentological and palaeobotanical changes across Moscovian and Kasimovian Euramerica, suggested that increased tectonic activity in the Variscan orogeny altered drainage patterns in Euramerica, creating conditions less suitable for the lycopsid flora and driving the disappearance of the coal swamps.

The family-level diversity curves of Sahney et al.[36] indicated an increase in faunal provinciality following the Moscovian, which they attributed to the rainforest collapse. It was suggested that the habitat fragmentation (the separation of the coal swamps into rainforest islands) was responsible. In support of this hypothesis,

they observed that amniotes were less severely affected by the environmental changes at this time, a result they attributed to the amniotic egg and the impermeable skin giving them independence from water[36].

The dispersal patterns of amniotes and amphibians identified by this study might appear to support the conclusions of Sahney et al.[36]. Both amphibians and amniotes exhibit a fall in dispersal rate and a peak in vicariance rate during the latest Carboniferous. The differing patterns observed in amphibians and amniotes are also supported here. The reduction of dispersal rate in amphibians is more substantial, indicating a greater sensitivity to the changing climate. Moreover, although the aridification trends continued throughout the Permian[28,31,37], the dispersal rates of amniotes recovered while those of amphibians remained low. It might appear that amniotes were more efficient at dispersal in the drier, more open habitat than the amphibians.

Before assuming that the results of this study emphatically support the conclusions of Sahney et al.[36], two caveats must be noted. First, it is not entirely clear to what extent the two are

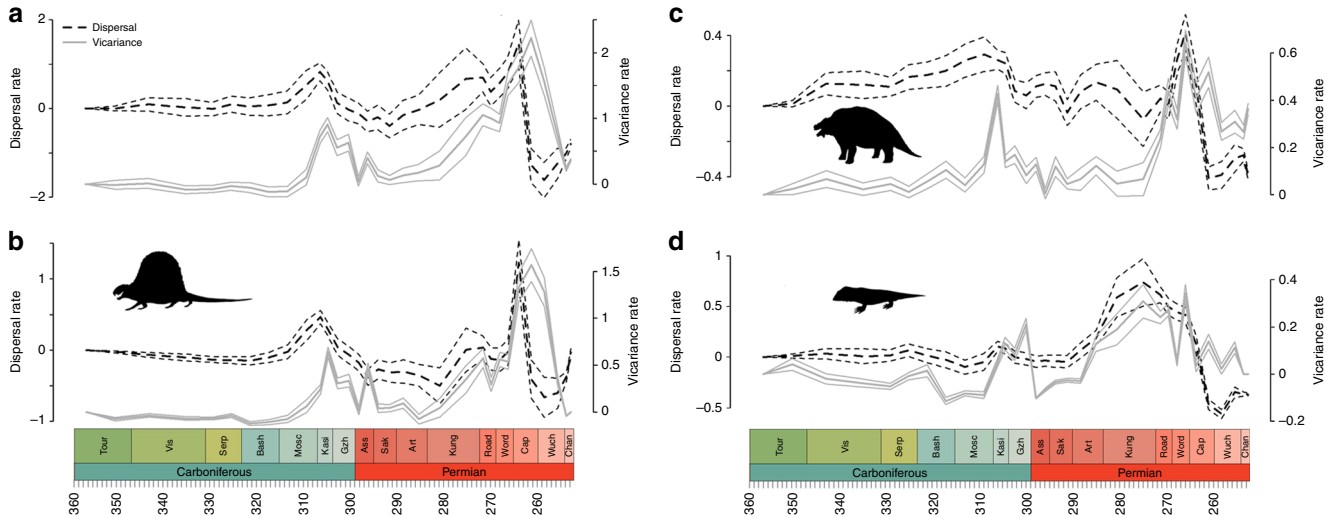

**Fig. 3** Dispersal and vicariance rates of amniote clades through time. **a** All Amniota; **b** Synapsida; **c** Parareptilia; **d** Eureptilia. Solid grey lines represent vicariance. Dashed black lines represent dispersal. Thick lines represent the mean of the rates obtained from each stochastic mapping iteration. The thinner lines represent a standard error above and below. Silhouettes from phylopic.org. **b** *Dimetrodon*, by Dmitry Bogdanov, available under Creative Commons Attribution-ShareAlike 3.0 Unported license. **c** *Scutosaurus*, by Chris Jennings (vectorized by A. Verrière), available under Public Domain Dedication 1.0 license. **d** *Concordia*, by Steven Blackwood, available under Creative Commons Attribution-ShareAlike 3.0 Unported license

directly comparable, since they are examining biogeography at different scales; Sahney et al.[36] were comparing the degree of endemism between formations and basins, while the results presented here illustrate dispersal patterns between continental-scale areas. Thus, our results do not provide any information on the habitat fragmentation model put forwards by Sahney et al.[36], but instead illustrate the development of geographic or climate barriers between considerably larger regions. The second caveat to note is that the model of Sahney et al.[36] makes less sense when viewed in the context of the more recent data regarding the rainforest collapse. Rather than an extended period of habitat fragmentation into the rainforest islands, detailed examination of the plant and pollen records indicate that the lycopsid-dominated coal swamps had disappeared by the earliest Kasimovian, thereafter replaced by fern-dominated, more open environments[26]. Therefore, the period in which we should observe the increased endemism at the formation level is extremely short, restricted to the late Moscovian. By the Kasmovian, the habitats were more uniform and should not exhibit continued isolation and reduced dispersal between them.

Despite these caveats, it is possible that the rainforest collapse may still have been responsible for the continental-scale results illustrated here. As mentioned above, the collapse was not synchronous across the continents, rather representing a contraction and progressive western shift[26]. If the coal swamps disappeared in Europe prior to North America, there would be an environmental barrier between the two, potentially restricting dispersal and driving vicariance. This explanation, however, does not resolve the second issue described in the previous paragraph: why would the reduced dispersal of amphibians continue into the early Permian with no environmental heterogeneity to prevent it?

An alternative explanation that would mitigate this objection might be the increased tectonic activity at the Variscan orogeny occurring during the Moscovian. This tectonic activity caused substantial uplift and terrestrial deformation, continuing into the Permian[26,38–40]. This mountain-building phase would have generated a substantial physical barrier between the equatorial localities, restricting not only movement between Europe and North America, but also migration between Gondwana and Laurasia. Elsewhere during the Moscovian, further uplift

and mountain-building was occurring, creating further barriers[40]. These include the Uralides between Eastern Europe and Asia, and the Appalachides between North and South America.

A serious issue requiring discussion is the conflict between the results presented here and those found by Dunne et al.[41]. The latter employed a networking approach to assess the biogeographic connectedness of localities during the Carboniferous and Cisuralian, implementing a correction to account for the phylogenetic non-independence of the taxa. They identified an increase in biogeographic connectedness between the Carboniferous and the Cisuralian, implying greater similarity of the faunas in the early Permian. This would seem to indicate increased dispersal across the boundary, the opposite signal to that observed here.

There are a number of possible explanations for the conflicting results. One possibility might be the differing temporal resolution of the studies: Here, dispersal rates are calculated at the substage level, while the finest resolution at which Dunne et al.[41] calculated biogeographic connectedness was in three bins, each containing two or three stages. However, this seems unlikely to be the cause of the discrepancy. The Gzhelian-Sakmarian dispersal rates identified here are consistently lower than those from earlier in the Carboniferous, but this interval has increased biogeographic connectedness in the analysis of Dunne et al.[41].

Another possible explanation is that the difference represents the different selection of regional divisions. Dunne et al.[41] defined their bioregions using cluster analysis of palaeocoordinates of localities, and so both North America and Europe were divided into numerous bioregions. Here, North America was divided into only two regions divided by the Hueco Seaway, and Western Europe was treated as a single region. The larger areas employed in this study means that dispersal between smaller subregions would not be counted. In order to test this possibility, the BioGeoBEARS analysis was repeated, using the same regions employed by Dunne et al.[41]. Since their study defines new bioregions following the Carboniferous-Permian boundary, a procedure not permitted by BioGeoBEARS, the supertree was time sliced to the end of the Carboniferous. This does not preclude the possibility of assessing the different results, as Dunne et al. noted an increase in biogeographic connectedness during the Gzhelian, which may still be tested.

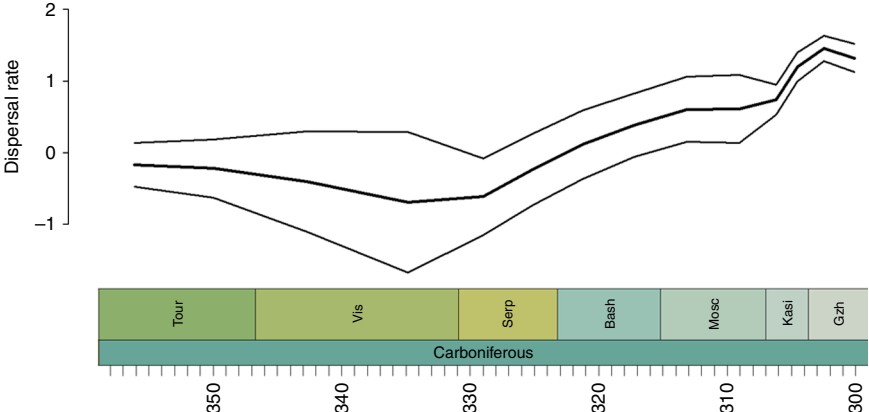

**Fig. 4** Carboniferous dispersal rates of all Tetrapoda, using the regional divisions of Dunne et al.[41]. Thick lines represent the mean of the rates obtained from each stochastic mapping iteration. The thinner lines represent a standard error above and below

This last analysis provided results that were more consistent with those obtained by Dunne et al.[41] (Fig. 4). Dispersal rates were low from the Tournasian until the Serpukovian but rose throughout the latter stages of the Carboniferous, reaching a peak in the Gzhelian. That this result is obtained when both North America and Europe are divided into multiple subregions indicates that different biogeographic patterns are occurring at different scales. Dispersal between the smaller-scale regions appears to become easier towards the end of the Carboniferous, although between the larger regions it becomes more difficult. This supports the inference that the principal barriers to dispersal in the late Carboniferous were the physical barriers between continental-scale regions rather than the local environmental barriers. The island-biogeography effect posited by Sahney et al.[36], with small, local islands of forest biogeographically isolated by open areas, is firmly rejected: dispersal between these local areas increased during the Gzhelian.

**Late Permian dispersal patterns**. While amphibian dispersal appears to have been restricted throughout the Permian, amniote dispersal increased throughout most of the Cisuralian. This has been noted previously; while the late Permian equatorial faunas contain amphibians characteristic of early Permian equatorial localities, the amniotes are more similar to those found in contemporary temperate regions[18,20]. Interestingly, the eureptiles show the same increase in dispersal as is found in synapsids. This contradicts previous suggestions that Captorhinidae (the most diverse eureptile family during the Permian) were better suited to arid environments[42]. It is only in the equatorial regions that this clade is abundant during much of the Permian; they do not become abundant in the temperate regions until the Lopingian[20]. However, while remaining rare in temperate regions, they were no less capable of dispersal than synapsids.

The first substantial decline in amniote dispersal does not occur until the late Capitanian, coinciding with a peak in vicariance rate. No significant change is observed in amphibians (perhaps simply due to them already being highly provincial).

The causes of this sudden and massive reduction of movement in amniotes is unclear, and our understanding is hampered by the lack of information regarding climate and environmental changes at this time. Environmental upheaval perhaps linked to flood volcanism has been suggested[43–45], as has relative climatic stability throughout the Guadalupian and Lopingian[46]. The nature and extent of the changes, and particularly their effect on terrestrial environments are particularly unclear[45,47]. Data from plants is contradictory, with some suggestion of a

substantial extinction at this time[48], but other suggestions that no substantial change in either diversity or floral composition occurred[34,35]. It seems unlikely that the lack of inferred dispersal events is due to geographically restricted sampling: the Wuchiapingian and Changshingian are among the few Paleozoic time bins where data is known from both palaeotemperate and equatorial latitudes in both Laurasia and Gondwana[20]. In fact, these stages contain a geographically wider sample than any other, with tetrapods known from all biogeographic regions under study except North America and northern South America.

A recent study using isotope data from amniote bones from the Karoo provides an indication of environmental changes in the temperate regions during the Permian[49]. The late Capitanian peak in $\delta^{13}C$ values was interpreted as an increase in water stress in plants i.e a period of substantially reduced humidity. By contrast, examinations of the geology and flora of contemporary palaeoequatorial formations in northern Africa and Europe found that the late Capitanian and early Wuchiapingian represented a wet phase interrupting the general trend towards aridification through the Permian[31,50]. It is possible that the contrasting trends in climate change at the different latitudes restricted the possibility of dispersal between the two.

**Conclusions**. Past discussions of biogeographic patterns within the Paleozoic Pangaea, and the extent of dispersal or vicariance across the continent, have been hindered by the limited use of quantitative methodology in making historical biogeographic inferences. As such, there has been considerable disagreement regarding the provinciality and cosmopolitanism of the tetrapod faunas, as well as the potential dispersal routes across Pangaea.

Quantitative analysis provides evidence of the existence of barriers to dispersal across Pangaea, and the timing of decreases in dispersal rates and peaks in vicariance rate provide evidence of their development. Two episodes of reduced dispersal are observed: in the late Carboniferous in amphibians and at the end of the Guadalupian in amniotes. Both dispersal troughs are accompanied by vicariance peaks. Although the nature of the new barriers must remain speculative for the present, the first shift in dispersal rate is attributed to increased mountain-building at that time, while the second is hypothesised to be due to climate barriers.

To conclude, it is important to note that historical biogeographic analyses not only provide information on the evolutionary history of the group under study, but also provide an additional line of evidence regarding broader patterns of earth's history. Where more usual lines of evidence regarding continental

arrangements or the development and break down of dispersal routes e.g. the Pangaea B scenario, the Carthaysian Bridge as a route from Gondwana to Laurasia, the relationships of the different faunas to each other provide a vital line of evidence.

## Methods

**Supertree construction.** The basis for the biogeographic analysis was a formal supertree of early tetrapods, designed to maximise taxonomic inclusivity. The full list of 49 source trees, finalised in April 2017, and the criteria for their inclusion is present in Supplementary Data 1 and Supplementary Note 1, respectively. The source trees were combined using the method "Matrix Representation with Parsimony" (MRP)[51,52]. Under this method, each node in each source tree is treated as a character. Taxa within a particular node in a particular source tree receive a score of "1" for the relevant character, taxa outside it receive a score of "0", and taxa not included in that source tree are scored as "?". The resulting matrix may be analysed using parsimony. The MRP matrix was generated using the programme Supertree0.85b[53]. The matrix was analysed in the Willi Hennig Society edition of TNT[54], using the driven search at level 100. The most parsimonious trees were searched for 100 times, and then a branch-and-bound search was carried out using each most parsimonious tree previously found as a start. The final supertree (the strict consensus) contains 593 terminal taxa, and is available in Supplementary Data 2. A summary version is presented in Fig. 5. Most of the terminal taxa are genera, but they also include some as-yet unnamed specimens and some species-level relationships where these have been tested. Clades which are not known from the fossil record until after the Permian but have ghost lineages extending into the Paleozoic were kept in the tree during time calibration but were dropped during the biogeographic analyses.

The supertree was time calibrated using the method of Lloyd et al.[55]. This method was itself based on an approach by Hedman[56], whereby the observed age of a node relative to its successive outgroups could be used to make inferences about sampling, and thereby assess how far back in time the node should be extended. Lloyd et al.[55] modified this method to be applied to an entire phylogeny. The tree was time calibrated in R 3.3.2[57]. In order to date the root node, stratigraphically consistent outgroups were required. In this case, the Devonian tetrapods *Ymeria*, *Ichthyostega* and *Metaxygnathus* were used. A maximum age constraint was placed on the root of 409.4 million years: the age of the split between tetrapods and lungfish inferred by a recent molecular clock study[58]. To account for the uncertainty surrounding the relationships and ages of taxa, 100 time calibrated trees were generated. For each of these, polytomies in the tree were randomly resolved and age ranges of each taxon were drawn from a uniform probability distribution covering the full possible range in which the taxon could have lived. The age ranges employed are shown in Supplementary Data 3. The analyses described below are carried out on all 100 trees.

**BioGeoBEARS.** To infer the timing and phylogenetic position of dispersal, vicariance and local extinction events, ancestral geographic ranges were inferred using a likelihood-based model comparison in the R package BioGeoBEARS[59]. The functions in this package implement three different biogeographic models: DEC, DIVA and BayArea[60,61], each of these allowing a different combination and geographic extent of biogeographic events. The phylogeny and the geographic ranges of the tips were analysed using all three models, with the Akaike information criterion used to identify the best-fitting model. This was found to be the DIVA model (Supplementary Table 1), which allows dispersal and local extinction along lineages, sympatric speciation of a lineage within a single area (but not sympatric speciation covering multiple region), and the vicariant origin of a species over a variety of range sizes.

The tetrapod-bearing formations of the Carboniferous and Permian were grouped into 13 bioregions. These were separated by a combination of physical barriers such as mountain ranges and internal seaways, and latitudinal lines intended to reflect climatic boundaries (see Supplementary Note 2 for detailed descriptions of the regions). The regions are: western North America; eastern North America, northern South America, southern South America, western Europe, eastern Europe, eastern Asia, northern Africa, southern Africa, Madagascar, India, Australia and Antarctica (see Supplementary Data 4 for the areas to which each taxon was assigned).

**Treefitting and the area-adjacency matrix.** The biogeographic analyses carried out in BioGeoBEARS incorporate a matrix indicating which areas are adjacent to each other, thus informing the algorithm about possible dispersal routes. While the majority of the area-adjacency matrix could be inferred with little debate, there are contentious issues which will influence it, for example: (1) whether the Pangaea B scenario, whereby North America is positioned more westerly than usually depicted and western Europe is adjacent to Northern South America[11,62], should be followed; (2) whether dispersal should be allowed along the Carthaysian Bridge[63,64] between eastern Gondwana (India and Australia) and Eastern Laurasia (East Asia); (3) whether it is necessary to treat the North American provinces on either side of the Hueco seaway[65] as separate biogeographic regions.

In order to answer such disputes, the supertree was subjected to a treefitting analysis. This analysis uses a phylogeny and a series of event costs to calculate the optimal biogeographic reconstruction (area cladogram) with the combination of events with minimum cost.

The treefitting analysis was carried out in Treefitter 1.3B1[66] using the heuristic search. The event costs used are: vicariance = 0.01, sympatric speciation = 0.01, extinction = 1, dispersal = 2. This cost scheme, whereby the cost of vicariance and sympatric speciation are minimised, is designed to maximise the likelihood of finding phylogenetically conserved distribution patterns[66,67].

Upchurch and Hunn[68] recommended time slicing (analysing taxa in each time bin separately) when using co-phylogeny reconstruction to analyse biogeography since area relationships change through time as barriers break down and new barriers develop. While these authors carried out time slicing by dropping all tips not present in a particular time slice, we modify this method by including tips with ghost lineages (unsampled portions of the fossil record that may be inferred from the phylogeny) extending into the time slice. The time slices tested were the Mississippian, Pennsylvanian, Cisuralian, Guadalupian and Lopingian. An area cladogram was calculated for each time slice (Supplementary Fig. 1).

The treefitting analyses were able to provide solutions to the three disputes discussed. In all area cladograms, the eastern North American region is more closely related to that of western Europe rather than western North America. It is therefore judged that the Hueco seaway as a significant barrier, while dispersal between eastern North America and western Europe was easier.

In none of the area cladograms does western Europe group with South America. While in most of the time bins the northern South American region is found with other Gondwanan regions, during the Mississippian it groups with western North America. These results argue strongly against the Pangaea B hypothesis; there is no evidence of close links between the faunas of South America and Europe.

Finally, there is strong evidence that the Cathaysian bridge has provided a link between the faunas of Gondwana and Laurasia. This is contra the suggestion of Cisneros et al.[11], who argued that, since no Paleozoic tetrapod fossils have been found in the regions of East Asia which then formed the Cathaysian Archipelago (South China and Korea), tetrapod dispersal between Gondwana and Laurasia took place via western Pangaea. However, in all area cladograms with the exception of the Lopingian time slice, East Asia is found to group more closely with Gondwanan faunas of Southern Africa than with the Eastern European fauna as argued by Cisneros et al.[11]. Thus, the lack of Paleozoic tetrapods in the Cathaysian regions is likely to be an artefact of preservation, and it is likely this was a frequent dispersal route between the South African and Eastern European regions, at least until the Guadalupian. The fact that in all Permian area cladograms the Eastern European region is found more closely related to the Gondwanan regions rather than western Europe would suggest that the Cathaysian Bridge was actually the preferred dispersal route between Gondwana and Laurasia, the Variscan Orogeny perhaps forming a more substantial barrier than suggested by Cisneros et al.[11]. The area-adjacency matrix inferred from these results and used in the BioGeoBEARS analysis is presented in Supplementary Table 2.

**Founder-event speciations.** Founder-event speciation, represented in BioGeoBEARS as speciation events where one of the two descendant lineages of a node remains in the ancestral area and the other jumps to a new area[59], was not permitted in the analyses in this paper. While it has been shown to be important in island clades[59], the model has been criticised for biasing towards cladogenetic dispersal events over anagenetic[69]. As the model incorporates no speciation-rate parameters, cladogenetic events like founder-event speciation are assumed to take place at fixed nodes, rather than being an outcome of a time-dependant diversification process. Anagenetic dispersal events, on the other hand, are treated as a character evolving in a stochastic manner similar to the MK model, and are therefore time-dependant. This difference increases the probabilities of inferring cladogenetic events such as founder-event speciation over anagenetic dispersal[59,69,70]. A further criticism was of the $j$ parameter, not as a rate of founder-speciation, but as a free parameter indicating the weight given to the easier mode of dispersal[69]. A final issue is that the founder events are also allowed to effectively bypass the area-adjacency matrix. While this is reasonable in island clusters, it produces unrealistic results a supercontinent where, short of circumnavigating the continent, the only possible dispersal routes for terrestrial animals are via adjacent areas.

**Stochastic mapping to infer dispersal and vicariance rates.** The BioGeoBEARS analysis identifies the probability that a node in the phylogeny was present in a particular area or combination of areas. One might infer the geographic ranges of the ancestral nodes by assuming this is represented by the most range with the highest probability (Figs. 6–8). However, a better alternative is to use these probabilities to infer possible biogeographic histories via a stochastic mapping approach: drawing ancestral regions using the probabilities inferred by the BioGeoBEARS analysis, and from these deducing the phylogenetic position and timing of biogeographic events required to obtain the observed biogeographic history.

For each of the 100 time calibrated phylogenies, 100 biogeographic histories were generated using stochastic mapping; thus 10,000 total evolutionary histories were tested. The timing of dispersal and vicariance events were extracted, and the number of each event in each time bin was counted. The time bins employed were informal substages, obtained by dividing each of the international stages in half. As

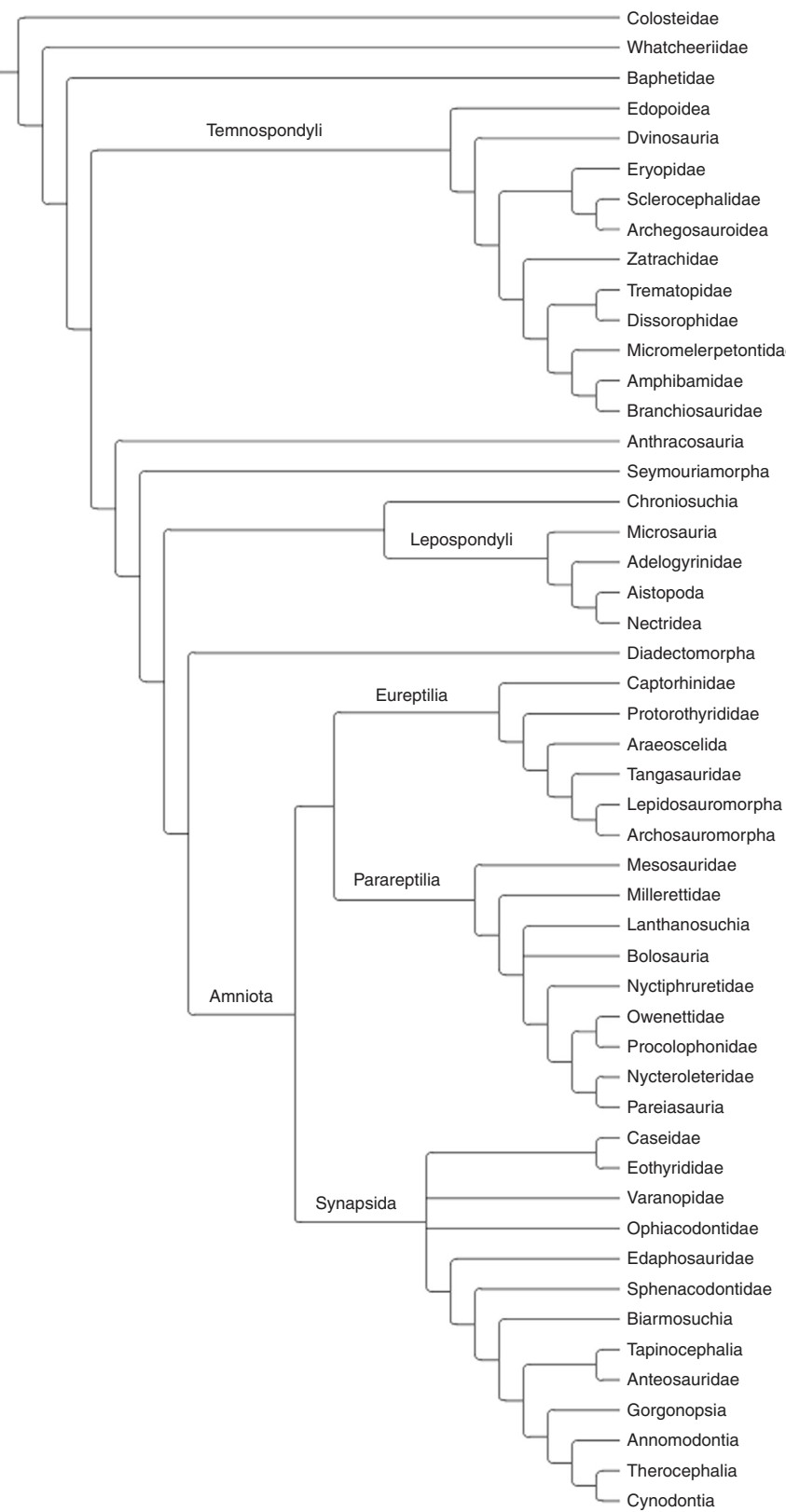

**Fig. 5** Summary version of the supertree used in the analysis

each bin differs in length, the event counts of each bin were divided by the length of the bin in order to obtain a rate.

The raw numbers of dispersal and vicariance events in each time bin may be misleading as a measure of dispersal or vicariance rate, due to the variation in the number of lineages and nodes in each time bin (Supplementary Figs. 2–6). If, for example, vicariance events were distributed at random across the nodes in the tree, time bins containing more nodes would exhibit a higher vicariance count simply by chance. Therefore, for each of the 10,000 stochastic maps, a null model was also generated by simulating 100 biogeographic histories over the time calibrated phylogeny. A starting area/range of areas was selected at random from the same set of regions used in the BioGeoBEARS analysis. With this starting area assigned to the root node of the relevant phylogeny, the

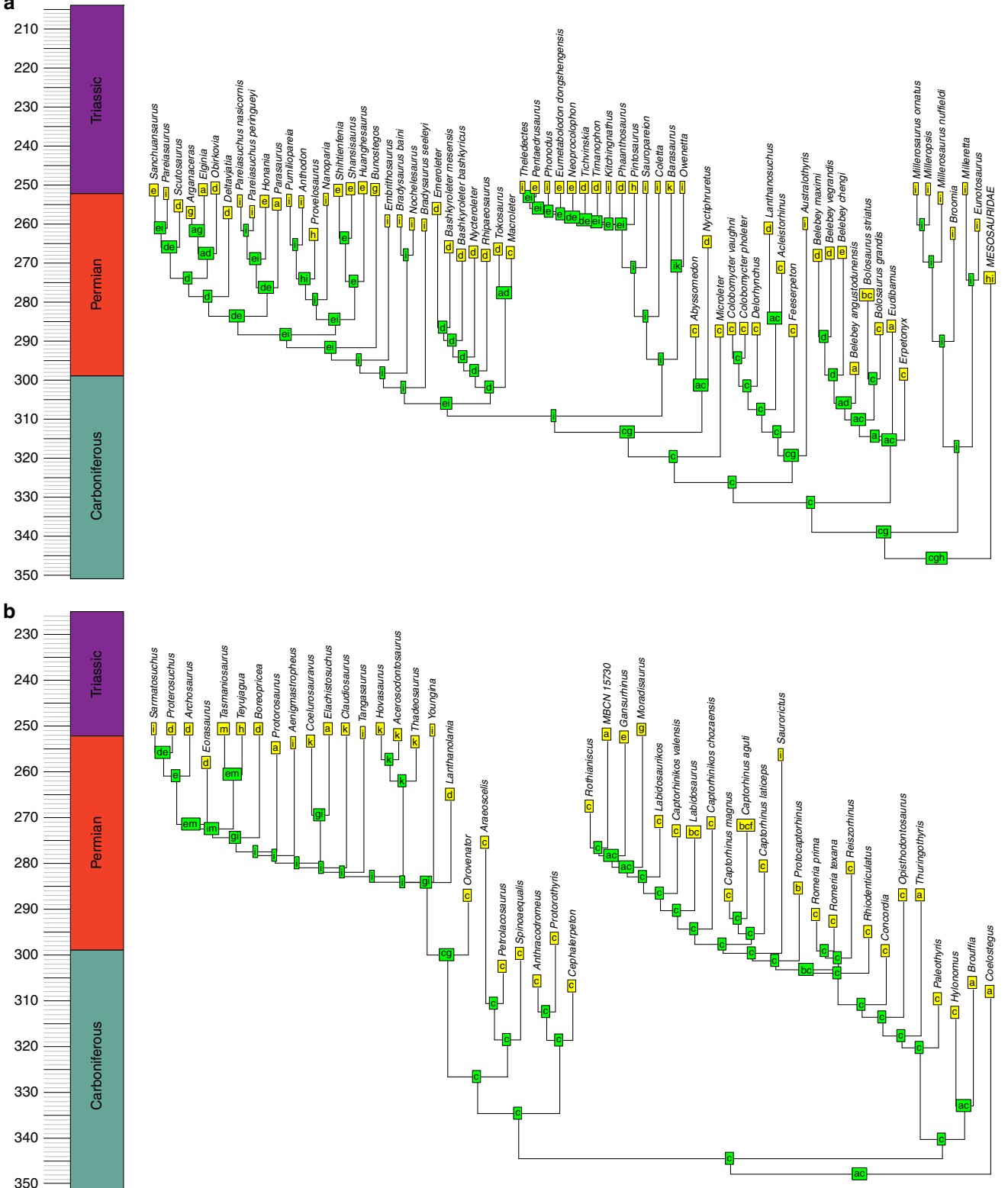

**Fig. 6** Example of a reconstruction of ancestral geographic ranges of sauropsids. Tree was randomly selected from the 100 time calibrated trees. **a** Parareptilia. **b** Eureptilia. Node labels represent the geographic range of that node with the highest probability, deduced by BioGeoBEARS. a = Western Europe; b = Western North America; c = Eastern North America; d = Eastern Europe; e = East Asia; f = Northern South America; g = Northern Africa; h = Southern South America; i = Southern Africa; j = Antarctica; k = Madagascar; l = India; m = Australia

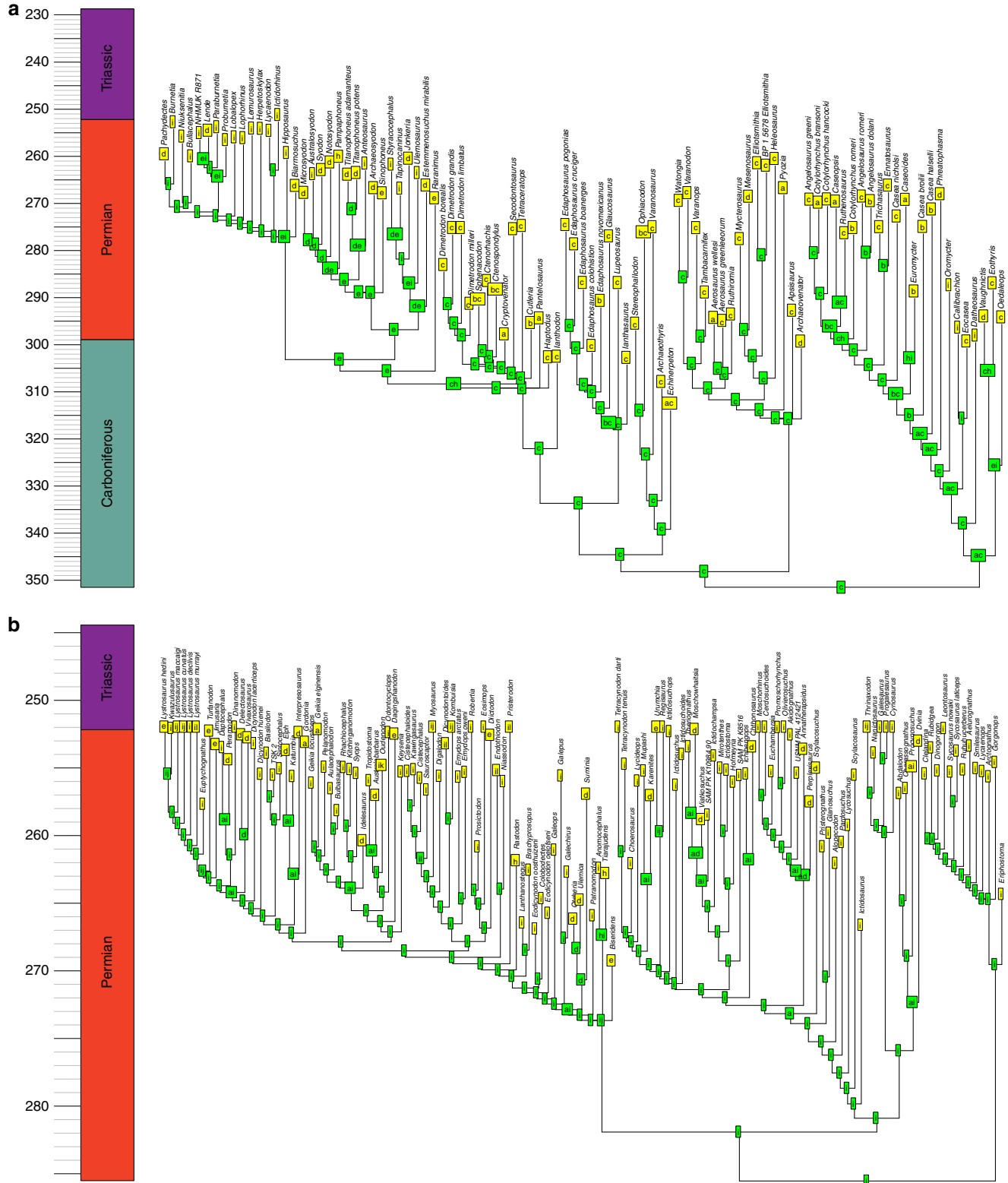

**Fig. 7** Example of a reconstruction of ancestral geographic ranges of Synapsids. Tree was randomly selected from the 100 time calibrated trees. **a** Pelycosaurian-grade synapsids, Biarmosuchia and Dinocephalia. **b** Neotherapsida. Node labels represent the geographic range of that node with the highest probability, deduced by BioGeoBEARS. a = Western Europe; b = Western North America; c = Eastern North America; d = Eastern Europe; e = East Asia; f = Northern South America; g = Northern Africa; h = Southern South America; i = Southern Africa; j = Antarctica; k = Madagascar; l = India; m = Australia

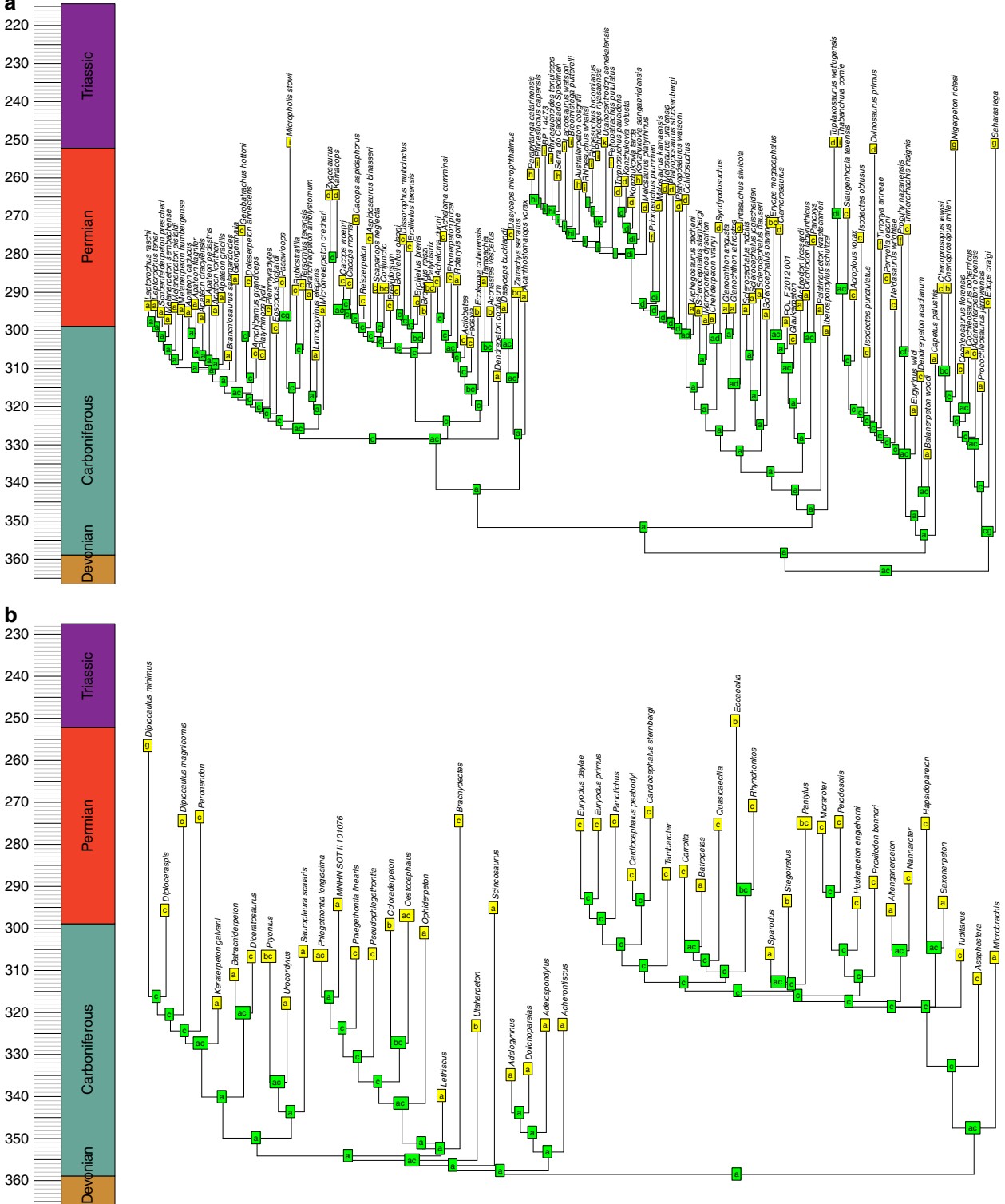

**Fig. 8** Example of a reconstruction of ancestral geographic ranges of amphibians. Tree was randomly selected from the 100 time calibrated trees.
**a** Temnospondyli. **b** Lepospondyli. Node labels represent the geographic range of that node with the highest probability, deduced by BioGeoBEARS.
a = Western Europe; b = Western North America; c = Eastern North America; d = Eastern Europe; e = East Asia; f = Northern South America;
g = Northern Africa; h = Southern South America; i = Southern Africa; j = Antarctica; k = Madagascar; l = India; m = Australia

biogeographic history would evolve stochastically, with the same biogeographic events permitted as were allowed by the DIVA model. The probabilities of dispersal, vicariance and local extinctions occurring at any point in time along a lineage/at a node taken from the results of the BioGeoBEARS analysis. The possible ranges of each node/lineage were constrained by the same parameters employed in the BioGeoBEARS analysis: the area-adjacency matrix limited the areas to which a lineage could disperse, and the regions employed could not be further subdivided i.e. a vicariance event could not occur at a node where the range covers only a single area. The R function to simulate the biogeographic history is provided in Supplementary Data 5.

Having simulated the 100 null histories and calculated the mean dispersal and vicariance rates of each, the null rates were subtracted from the observed rates. Thus, a negative dispersal/vicariance rate would indicate that the observed rate is less than would be expected given a randomly evolved biogeographic history over the same phylogeny.

**Code availability**. All custom code used in this study is available in Supplementary Data 5.

## Data availability

All data generated and analysed during this study are included in this published article (and its supplementary information files). A reporting summary for this article is available as a Supplementary Information file.

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

## Acknowledgements

We are grateful to Richard J. Butler and Graeme T. Lloyd for their helpful comments and discussion. The research of N.B. and J.F. was funded by Deutsche Forschungsgemeinschaft grant number FR 2457/5-1 awarded to J.F. The research of E.M.D. and D.D.C. was funded by the European Union's Horizon 2020 research and innovation programme 2014–2018 under grant agreement 637483 (ERC Starting Grant TERRA to Richard J. Butler).

## Author contributions

N.B., E.M.D. and D.D.C. collected data. N.B. designed and carried out analyses. N.B., E.M.D., D.D.C. and J.F. contributed to the writing of the manuscript.

## Additional information

**Competing interests:** The authors declare no competing interests.

