## [Peer Review File · Nature Communications]

Reviewers' Comments:

Reviewer #1:

Remarks to the Author:

This paper addresses an interesting, large-scale evolutionary topic appropriate for readers of Nature Communications. In the introduction, the authors provide a good background on the pattern of faunal changes during the Permo-Carboniferous and then propose a new method to explore what might have driven the observed patterns. I really like the introduction and agree whole-heartedly with the problems identified by the authors concerning incrementally more basal taxa depending on biogeographic hypotheses. So it was a disappointment when I found the methods section difficult to understand and, as such, I'm not in a great place to assess the validity of the results. As a result, all of my comments relate to the methods in some way.

Here's how I think the method works:

Gather hypothesis of phylogenetic relationships with observed geographic position of terminal taxa -> generate ancestral state reconstructions of geographic position for internal cladogram nodes -> analyze geographic changes between nodes (and between nodes and tips?) -> compare numbers of inferred geographic changes (vicariance and dispersal, based on adjacency matrix) to null distribution. If any of this is incorrect, please take a look at clarifying the relevant sections of the methods section.

Comment 1: The rationale for using 13 geographic regions wasn't sufficiently explained. At the heart of this paper (and several recent others) is finding a method to subdivide a single supercontinent into meaningful biogeographic units: bigger than individual fossil localities, but smaller than the entire supercontinent. From my reading of the methods section, the 13 regions selected are based partly on modern political units and partly on paleoenvironmental regions. The latter seems like an interesting, and potentially biologically meaningful approach, but the former seems like folly. What does "Northern South America" mean when it is continuous with the rest of South America, Africa, and North America? Is there a better way to quantify regions that doesn't rely on something so arbitrary? Geologic basins could be a useful approach. Or make location a continuous character instead of binning it into 13 regions.

Comment 2: How does inferring rates of dispersal and vicariance through time tell us about the processes generating those rates? There seems to be a big disconnect between what's analyzed and the conclusions drawn. What have the authors done to eliminate other factors that could influence the rates under investigation? It seems like the authors suggest climate (and tectonics) because they are the usual suspects, not because their analysis provides evidence for either. The authors should discuss other factors that could produce the results they present.

Comment 3: What is the rationale for not including error bars on the graphs in Figure 2? As seen in the supplementary data files, many of the analyses show wide error bars and I think readers should see them up front, so they can decide for themselves the significance of the changes described. For example, the error bars for Figure 2A (=Supplemental Figure 16) suggest no significant change for the vast majority of the time involved.

MINOR EDITS:

For each of the following, L# refers to the line numbers on the side of each page in the main document PDF.

L52: Change to read "By the late Permian, terrestrial..."

L55: Briefly identify the substantial environmental and climate changes here.

L58: Insert "relatively" or "strongly" before cosmopolitan.
L58: Previous researchers would probably suggest a cosmopolitan fauna at the family level (certainly not species). This is worth noting.
L60: Fix typo: biogeography
L62: Fix typo: tetrapods
L70: Insert "within a species" after populations.
L70: Replace "repeated patterns" with "shifting interpretations"
L81: Not clear what you mean by "lineages" – Is this a clade/node?
L120: Update to: "responses to climate change occurring"
L161: Fix typo: forward
L161: Replace "illustrate" with "support"
L178: Replace "which" with "that"
L178: Replace "account for" with "mitigate"
L183: Replace "movement" with "migration of tetrapods"
L246: Could this issue be caused by the Karro late Permian swamping out all other data?
L265: Fix typo: wet

Christian Sidor

Reviewer #2:

Remarks to the Author:

Overview

This is an important and interesting study that should have broad appeal. I found a few typos and other small errors. I have one point where greater clarification is needed with regard to the methodological approach (see last point below). Therefore, I think this can be accepted for publication subject to minor revisions.

Specific points

Lines 50-57 – There is a series of introductory statements, but with no references to back them up. Please add.

Line 60 - '...identified [10-19] there has been very little study into the historical biogeographic of...'
'biogeographic' should be 'biogeography'

Lines 132-134 – '...the equatorial latitudes, with macrofloral and palynological evidence suggesting coals development continued into the earliest Kasimovian in American localities [24], but after this the last...'

'coals' should be 'coal'

Lines 146-147 – 'The dispersal patterns of amniotes and amphibians identified by this study might appear support the conclusions of Sahney et al. [34].'

'appear support' should be 'appear to support'

Lines 151-153 – '...trends continued throughout the Permian [26,29,35], the dispersal rates of amniotes recovered while that of amphibians remained low.'

Can't have 'rates' for amniotes and then 'that' (i.e. singular rate) for amphibians – change 'that' to 'those'

Lines 153-156 - at dispersal in the drier, more open habitat than the amphibians. Before jumping to the conclusion that the results of this study emphatically support the conclusions of Sahney et al. [34], two caveats must be noted.'

'jumping to the conclusion' is a rather informal way of saying this - what about 'assuming that' And, having the word 'conclusion in the sentence twice is repetitive.

Line 219 - '...and Europe are divided into multiple subregions indicates that different biogeographic patterns...'

'biogeographic' should be 'biogeographic'

Line 262 - '...late Capitanian peak in $\delta^{13}\text{C}$ values was interpreted as an increase water stress in plants...'

'increase water' should be 'increase in water'

Lines 366-374 - when producing the null model for dispersal events,. Did this take into account the rules used for area absence, and so allowed/disallowed dispersal routes that were used in the BioGeoBEARS analyses? If so, please state this. If not, then this might not be a fair comparison with the events found by BioGeoBEARS.

Paul Upchurch

Reviewer #3:

Remarks to the Author:

Brocklehurst et al. re-examine Paleozoic tetrapod trends in dispersal and vicariance with recently developed biogeographic models. By simulating evolutionary histories under these models, their research introduces a framework to identify epochs with remarkably high/low numbers of dispersal and vicariance events. With this, Brocklehurst and colleagues identify that the dispersal and vicariance was generally more common in the Carboniferous than the Permian, which has implications for conceptual models of Late Paleozoic biogeography, specifically concerning the importance of ecological and geographical barriers to biogeography. In order to study this question, the authors assembled a sizable comparative biogeographic dataset for Paleozoic tetrapods, much of which is made available to the community through the Supp Info.

Overall, I think this is interesting work. The paper is written clearly and the study design is sound. The work is one of relatively few recent studies aimed at studying paleobiogeography using new models from the statistical phylogenetics literature. In this sense, it may serve as an example for future paleobiogeographic studies.

My review of the manuscript focuses primarily on the statistical aspects of the biogeographic analysis. I'm not as well-versed in the Paleozoic tetrapod biogeography literature, but I'll say that I found that the authors were appropriately cautious in interpreting their results, and generous when giving context to what their findings mean. Besides minor comments and suggestions, which are given at the bottom of this review, there are several major points to raise here:

Figure 1 should be replaced by a time-calibrated phylogeny with geological epochs on the x-axis and ancestral state estimates (or one stochastic mapping) on the phylogeny itself. This will not only help readers understand the inherent phylogenetic context of the biogeographic question, but it will also greatly aid the interpretation of Figure 2.

The dispersal and vicariance rates are given an unusual definition that is closer to "relative counts" than "rates". Proper event rates will likely serve better as a measurement, since they offer a meaning that is not fundamentally linked to the number of lineages present within a time bin. Event rates can be computed by taking the number of events within a time bin and divide that by the total branch length within that bin (\sim the product of the # of lineages by the bin width). My sense is this could potentially reduce some effects of heterogeneous stochasticity/sampling error across time bins. Null rates (rather than null counts) can be computed using the randomization test the authors proposed.

That said, the null model is appealing due to its simplicity. But one issue it has is that "too many" vicariance event may be sampled in close proximity (a narrow range cannot be subdivided multiple times without intermediate range expansions, which is not required by uniform random sampling). A more realistic null model would involve simulating datasets under the DEC parameters, fitting DEC to those data, then binning ancestral range estimates per usual. Doing this is more work, but it would yield a more meaningful null distribution of biogeographic events.

The authors should be aware that Dispersal-Extinction-Cladogenesis models ignore speciation events that leave no sampled descendants (due to extinction/poor fossilization/sampling). This means that the DEC vicariance counts are generally underestimates (but never overestimates) of the true number of vicariance events. Because the vicariance counts are "relative" to the null counts, this may or may not be an issue.

Thank you for the invitation to review this paper.

Signed,
Michael Landis

Minor comments:

60: "historical biogeographic" to "historical biogeography"

79--80: "A supertree ... (Fig. 1)" -- It's confusing that Figure 1 appears to show a backbone tree rather than a super tree.

85, 100, and elsewhere: Instances of "rates" should be replaced with "counts", or more preferably, "relative counts" or "excess counts". Alternatively, these quantities could be computed as actual rates rather than as relative counts (see main comment).

219: "biogeographic" to "biogeographic"

265: "west phase" or "wet phase"?

292: More emphasis could be placed on the size and scope of the tree. Simple information like how many taxa are in the tree? For those of us who work outside Permian tetrapod systematics, how many important subclades are represented?

324: Cite Ronquist (1997) in reference to DIVA and Landis et al (2013) in reference to BayArea.

337: Why not incorporate time-stratified geographic effects?

353--354: What is the time bin width?

355--359: "time bins containing more nodes would exhibit a higher vicariance rate simply by chance"
-- no, those time bins would exhibit a higher vicariance count, not a higher rate (number of events divided by number of nodes)

372-374: The text insinuates that perhaps only one dispersal event is allowed per branch ("the dispersal event", "a dispersal event", etc) -- but the null model should be able to place multiple dispersal events on a single branch.

Figure 2 could also contain a subfigure with the number of lineages represented by each clade vs. time (LTT plots). LTT plots would calibrate how surprised the reader might be to learn what clades do or do not match the null expectations. For example, Synapsids seem to be driving the signal for the reduced vicariance rate among Amniotes -- but is that because the phylogeny underrepresents Eureptiles and Parareptiles towards the Late Permian? And consider changing Figure 2F so the x-axis so it has the same width as Carboniferous for Figures 2A-E.

Supplementary Table 1 should give the parameter estimates. Does mean likelihood refer to the mean over 100 tree samples?

References

Ronquist, F. (1997). Dispersal-vicariance analysis: a new approach to the quantification of historical biogeography. *Systematic Biology*, 46(1), 195-203.

Landis, M. J., Matzke, N. J., Moore, B. R., & Huelsenbeck, J. P. (2013). Bayesian analysis of biogeography when the number of areas is large. *Systematic Biology*, 62(6), 789-804.

Reviewer 1

“I found the methods section difficult to understand... Here’s how I think the method works: Gather hypothesis of phylogenetic relationships with observed geographic position of terminal taxa-> generate ancestral state reconstructions of geographic position for internal cladogram nodes-> analyze geographic changes between nodes (and between nodes and tips?) -> compare numbers of inferred geographic changes (vicariance and dispersal, based on adjacency matrix) to null distribution. If any of this is incorrect, please take a look at clarifying the relevant sections of the methods section.”

- The reviewer is correct in that this is the methodological procedure. We have attempted to describe it more clearly in the text

“The rationale for using 13 geographic regions wasn’t sufficiently explained. At the heart of this paper (and several recent others) is finding a method to subdivide a single supercontinent into meaningful biogeographic units: bigger than individual fossil localities, but smaller than the entire supercontinent. From my reading of the methods section, the 13 regions selected are based partly on modern political units and partly on paleoenvironmental regions. The latter seems like an interesting, and potentially biologically meaningful approach, but the former seems like folly. What does “Northern South America” mean when it is continuous with the rest of South America, Africa, and North America?”

- The 13 geographic regions and the rationale behind their definition have been described in considerably greater detail than previously in the Supplementary methods.
- We would like to emphasise that political units were not used in defining the bioregions; they were given “political” names purely to help readers not so familiar with the Paleozoic tetrapod-bearing basins understand the divisions. The regions are defined based on a combination of mountain ranges, internal seaways and latitude (the latter an attempt to incorporate potential climatic barriers).

“Is there a better way to quantify regions that doesn’t rely on something so arbitrary? Geologic basins could be a useful approach. Or make location a continuous character instead of binning it into 13 regions.”

- As mentioned above, the bioregions are not defined arbitrarily, but are based on a combination of potential barriers to dispersal.
- It is worth noting that, due to the geographically patchy record of Paleozoic tetrapods, the separation of the bioregions in this way usually ended up simply separating the individual basins. To cite the reviewer's own example of the definition of Northern vs Southern South America, the intention was to separate them by latitude (palaeoequatorial vs palaeotemperate), but the result was a separation of the Paraná and Parnaíba basins.
- We chose not to treat location as a continuous character as we do not feel this is an appropriate way of analysing biogeography. A paper detailing our justification of this attitude is in preparation, but a brief summary of the reasons is as follows:
 - Methods of studying biogeography using continuous geographic data generally treat dispersal as a Brownian motion "walk" of lineages across geospace. This does not represent well the biogeographic processes by which organisms disperse, ignoring range expansions and peripheral speciation.
 - Such methods have yet to produce a reliable way of studying vicariance.
 - The division of geospace into discrete areas bounded by physical and environmental barriers better represents the way organisms map themselves onto geospace.

"Comment 3: What is the rationale for not including error bars on the graphs in Figure 2? As seen in the supplementary data files, many of the analyses show wide error bars and I think readers should see them up front, so they can decide for themselves the significance of the changes described."

- This decision was purely aesthetic; many of the graphs were quite crowded and so adding the error margins made the curves difficult to distinguish. The figure has now been split into multiple figures so that less curves can be shown in each graph and the error margins can be added.

Reviewer 2

"Lines 50-57 – There is a series of introductory statements, but with no references to back them up. Please add."

- This has been done

"Lines 366-374 – when producing the null model for dispersal events,. Did this take into account the rules used for area absence, and so allowed/disallowed dispersal routes that were used in the BioGeoBEARS analyses? If so, please state this. If not, then this might not be a fair comparison with the events found by BioGeoBEARS."

- In the new model used in this version, all the same constraints and parameters used in the BioGeoBEARS analysis were carried over into the null model, including the area adjacency.

Reviewer 3

“Figure 1 should be replaced by a time-calibrated phylogeny with geological epochs on the x-axis and ancestral state estimates (or one stochastic mapping) on the phylogeny itself. This will not only help readers understand the inherent phylogenetic context of the biogeographic question, but it will also greatly aid the interpretation of Figure 2.”

- We have added figures showing the ancestral state estimates over one randomly selected time calibrated phylogeny. As there are 594 taxa it was not possible to show all the states legibly on one tree, so the five major clades are shown in three separate figures. Figure 1 is retained to show the relationships of these clades to each other.

“The dispersal and vicariance rates are given an unusual definition that is closer to "relative counts" than "rates". Proper event rates will likely serve better as a measurement, since they offer a meaning that is not fundamentally linked to the number of lineages present within a time bin. Event rates can be computed by taking the number of events within a time bin and divide that by the total branch length within that bin (~ the product of the # of lineages by the bin width). My sense is this could potentially reduce some effects of heterogeneous stochasticity/sampling error across time bins.”

- The observed and null dispersal and vicariance counts have been converted to rates by dividing them both by bin length instead of by total branch length in the bin (the reason for this is that the null models are intended to account account for branch length and number of nodes within each bin).

“That said, the null model is appealing due to it's simplicity. But one issue it has is that "too many" vicariance event may be sampled in close proximity (a narrow range cannot be subdivided multiple times without intermediate range expansions, which is not required by uniform random sampling). A more realistic null model would involve simulating datasets under the DEC parameters, fitting DEC to those data, then binning ancestral range estimates per usual. Doing this is more work, but it would yield a more meaningful null distribution of biogeographic events.”

- We agree with the reviewer’s criticism and are grateful for his suggestion. We have revised the null model, generating it by stochastically evolving biogeographic histories with the same parameters as the empirical analysis instead of distributing the events at random. The dispersal and vicariance curves do show some differences to the original results and so we have altered the text in the appropriate places. The two events that form the bulk of our discussion are still observed, and so our main conclusions are not changed.
- It should be noted that it was the DIVA, not DEC, model which was found to best fit our data, so the null biogeographic histories were generated using this model

“292: More emphasis could be placed on the size and scope of the tree. Simple information like how many taxa are in the tree? For those of us who work outside Permian tetrapod systematics, how many important subclades are represented?”

- More data on the supertree and source trees have been added

“353--354: What is the time bin width?”

- The time bins are created by dividing the international stages into two, early and late. This has now been specified.

“372-374: The text insinuates that perhaps only one dispersal event is allowed per branch ("the dispersal event", "a dispersal event", etc) -- but the null model should be able to place multiple dispersal events on a single branch.”

- The null model does allow multiple dispersal events along each branch. This has been clarified.

“Figure 2 could also contain a subfigure with the number of lineages represented by each clade vs. time (LTT plots). LTT plots would calibrate how surprised the reader might be to learn what clades do or do not match the null expectations. For example, Synapsids seem to be driving the signal for the reduced vicariance rate among Amniotes -- but is that because the phylogeny underrepresents Eureptiles and Parareptiles towards the Late Permian?”

- These have been added to the supplementary material (main-text figures are approaching the limit for this journal)
- The reviewer is correct as to why the synapsid signal is largely driving the amniote signal, although one perhaps should not say that eureptiles and parareptiles are underrepresents: they were considerably less diverse throughout the Permian than synapsids. In the same way, the Carboniferous tetrapod signal is largely driven by the amphibians, but after amniotes radiate, they dominate the Permian signal.

“Supplementary Table 1 should give the parameter estimates. Does mean likelihood refer to the mean over 100 tree samples?”

- This has been added.
- The mean is over the 100 trees. This has been clarified in the caption

Reviewers' Comments:

Reviewer #1:

Remarks to the Author:

I have reviewed the revised manuscript and have no additional suggestions. It seems ready for publication.

Reviewer #3:

Remarks to the Author:

Thank you for inviting me to review the new revisions. I was pleased to see the improved method for estimating null rates of biogeographic events, which relies on a simulation script shared by the authors. The script's simulating model had some properties that were unusual for models in the DEC family. Hopefully, the authors can clarify those issues. If the issues are stand, then it may (unfortunately) require that the null rate distributions be recomputed.

Signed,
Michael Landis

Line 180:

"One might be" -- missing "possibility"??

Figures 2--4:

These new figures of phylogenies with times and ancestral areas are extremely useful. There is a minor numbering error, where the captions appear as Fig 2, 2, 3 instead of 2, 3, 4.

Lines 567-583:

Using simulations to approximate the null rate of dispersal/extirpation/vicariance over time is an improvement to the previous null rate estimate. However, the authors should be aware that the simulation may not behave as intended. For example, the expected number of events over an interval of time should equal the product of the event rate and the time duration. At a minimum, this should be modeled using the Poisson distribution.

```
> set.seed(1)
> branch.length <- 10
> disp.rate <- 0.3
>
> # Poisson distribution
> # Expected number of events, E[N] = branch.length * disp.rate
> branch.length * disp.rate
[1] 3
> # Correct distribution for number of simulated events
> no.disp.correct <- rpois(n=1E4, lambda=branch.length*disp.rate)
> mean(no.disp.correct)
[1] 3.0051
> var(no.disp.correct)
[1] 3.040178
>
```

```

> # Simulation code
> disp.test<-runif(1E4,0,1)
> no.disp<-round((branch.length * disp.rate )/disp.test)
> # Mean number of simulated events
> mean( no.disp )
[1] 26.8695
>
> # Variance number of simulated events
> var( no.disp )
[1] 59786.97
>

```

Because the mean and variance are equal under the Poisson distribution, the discrepancy does not seem to be a simple scaling artifact of rate or branch.length.

The biogeography simulator also appears to assume the dispersal and extirpation rates are independent of range size. Under DEC's anagenetic process, the rate of a range losing *_any_* area is the sum of extirpation rates across areas (the rate of losing any area from the range ABC is $e_A+e_B+e_C$ or $3*e$ when all extirpation rates are equal across areas). Likewise, the rate of a range of gaining *_any_* new area is the sum of dispersal rates from currently occupied areas into the new area. Both of these rates depend on the range size, which changes over time, and thus a simple Poisson model won't hack it. A continuous-time Markov chain is needed.

This is to say that the null model simulations are not really under the DEC inference model, and so the simulations don't give us a set of strictly comparable evolutionary histories. At a minimum, simulating the number of events should follow a simple Poisson distribution (example above). If data is simulated this way, then the text must be explicit that the simulator is not an exact DEC model. Alternatively, a full DEC simulator could be used to test the null model. I am not sure whether BioGeoBEARS or LAGRANGE have simulators, but Nick Matzke and Rick Ree would be able to help there. RevBayes does have a simulator that could be useful: https://revbayes.github.io/tutorials/biogeo/biogeo_simple.html

At this point, it's good to remember that the simulated null counts are eventually converted into null rates. This lets the authors identify time bins of interest when "too many" or "too few" biogeographic events are inferred for the real dataset relative to the simulations. In this light, I can't convince myself that the empirical results should hold regardless of the design of the simulator. If the simulated event counts/branch are not what the authors intended, this would require another round of simulations.

Response to Reviewers:

We have resubmitted the manuscript entitled “**Physical and environmental drivers of Palaeozoic tetrapod dispersal across Pangaea**”, having made the changes in methodology and text suggested by reviewer 3 (the only reviewer who had comments on this version. This includes a further change to the simulation procedure used to generate null rates of dispersal and vicariance, incorporating continuous time Markov chains. The new simulations caused only minor changes in the results, and the text has been changed in the appropriate places to include these. The two events which make up the bulk of our discussion are still observed, and so our main conclusions are unchanged

We here provide a response to the reviewer’s comments, indicating where changes have been made. Both minor comments relating to wording, and errors in figure numbering have been changed according to reviewer’s suggestions.

Reviewer #1 (Remarks to the Author):

I have reviewed the revised manuscript and have no additional suggestions. It seems ready for publication.

Reviewer #3 (Remarks to the Author):

Thank you for inviting me to review the new revisions. I was pleased to see the improved method for estimating null rates of biogeographic events, which relies on a simulation script shared by the authors. The script's simulating model had some properties that were unusual for models in the DEC family. Hopefully, the authors can clarify those issues. If the issues are stand, then it may (unfortunately) require that the null rate distributions be recomputed.

Signed,
Michael Landis

Line 180:
"One might be" -- missing "possibility"??

Figures 2--4:
These new figures of phylogenies with times and ancestral areas are extremely useful. There is a minor numbering error, where the captions appear as Fig 2, 2, 3 instead of 2, 3, 4.

Lines 567-583:
Using simulations to approximate the null rate of dispersal/extirpation/vicariance over time is an improvement to the previous null rate estimate. However, the authors should be aware that the simulation may not behave as intended. For example, the expected number of events over an interval of time should equal the product of the event rate and the time duration. At a minimum, this should be modeled using the Poisson distribution.

```
> set.seed(1)
> branch.length <- 10
> disp.rate <- 0.3
>
> # Poisson distribution
```

```

> # Expected number of events, E[N] = branch.length * disp.rate
> branch.length * disp.rate
[1] 3
> # Correct distribution for number of simulated events
> no.disp.correct <- rpois(n=1E4, lambda=branch.length*disp.rate)
> mean(no.disp.correct)
[1] 3.0051
> var(no.disp.correct)
[1] 3.040178
>
> # Simulation code
> disp.test<-runif(1E4,0,1)
> no.disp<-round((branch.length * disp.rate )/disp.test)
> # Mean number of simulated events
> mean( no.disp )
[1] 26.8695
>
> # Variance number of simulated events
> var( no.disp )
[1] 59786.97
>

```

Because the mean and variance are equal under the Poisson distribution, the discrepancy does not seem to be a simple scaling artifact of rate or branch.length.

The biogeography simulator also appears to assume the dispersal and extirpation rates are independent of range size. Under DEC's anagenetic process, the rate of a range losing *any* area is the sum of extirpation rates across areas (the rate of losing any area from the range ABC is $e_A+e_B+e_C$ or $3*e$ when all extirpation rates are equal across areas). Likewise, the rate of a range of gaining *any* new area is the sum of dispersal rates from currently occupied areas into the new area. Both of these rates depend on the range size, which changes over time, and thus a simple Poisson model won't hack it. A continuous-time Markov chain is needed.

This is to say that the null model simulations are not really under the DEC inference model, and so the simulations don't give us a set of strictly comparable evolutionary histories. At a minimum, simulating the number of events should follow a simple Poisson distribution (example above). If data is simulated this way, then the text must be explicit that the simulator is not an exact DEC model. Alternatively, a full DEC simulator could be used to test the null model. I am not sure whether BioGeoBEARS or LAGRANGE have simulators, but Nick Matzke and Rick Ree would be able to help there. RevBayes does have a simulator that could be useful:

https://revbayes.github.io/tutorials/biogeno/biogeno_simple.html

At this point, it's good to remember that the simulated null counts are eventually converted into null rates. This lets the authors identify time bins of interest when "too many" or "too few" biogeographic events are inferred for the real dataset relative to the simulations. In this light, I can't convince myself that the empirical results should hold regardless of the design of the simulator. If the simulated event counts/branch are not what the authors intended, this would require another round of simulations.

- I am grateful to the reviewer for pointing out this mistake; I misunderstood the nature of the d and e parameters from the papers describing BioGeoBEARS. I have therefore adopted his second suggestion and now simulate dispersal and local extinction under a continuous time Markov process using functions from the R package `spuRs`. The Q matrix used in this simulation is calculated as described in Ree and Sanmartin's 2008 paper describing the DEC model. Vicariance is still simulated as before, since it is not a process with a probability of occurring at any point in a continuous time range, but

is instead a process with a probability of occurring at a specific point in time (the age of the node).

- The new simulations did include some slight changes to the results:
 - A new slight peak in vicariance in the Serpukhovian
 - The fact that the Carboniferous decline in dispersal is visible both in temnospondyles and lepospondyles instead of just the former
 - The Capitanian decrease in dispersal is now seen in temnospondyles instead of just in amniotes.
- However, these changes do not affect our interpretations or conclusions.

Reviewers' Comments:

Reviewer #3:

Remarks to the Author:

I appreciate that the authors made the effort to correct the "null model" results. Good work. It's fortunate that the results were not significantly impacted. I have no further comments to make for this review.

Signed,

Michael Landis